# Musical and Non-Musical Sounds Influence the Flavour Perception of Chocolate Ice Cream and Emotional Responses

**DOI:** 10.3390/foods11121784

**Published:** 2022-06-17

**Authors:** Yi Hsuan Tiffany Lin, Nazimah Hamid, Daniel Shepherd, Kevin Kantono, Charles Spence

**Affiliations:** 1Department of Food Science, Auckland University of Technology, Private Bag 92006, Auckland 1142, New Zealand; tlin@aut.ac.nz (Y.H.T.L.); kkantono@aut.ac.nz (K.K.); 2Department of Psychology, Auckland University of Technology, Private Bag 92006, Auckland 1142, New Zealand; daniel.shepherd@aut.ac.nz; 3Crossmodal Research Laboratory, Department of Experimental Psychology, New Radcliffe House, University of Oxford, Oxford OX2 6GG, UK; charles.spence@psy.ox.ac.uk

**Keywords:** chocolate ice cream, music, sound, temporal dominance of sensations, emotion

## Abstract

Auditory cues, such as real-world sounds or music, influence how we perceive food. The main aim of the present study was to investigate the influence of negatively and positively valenced mixtures of musical and non-musical sounds on the affective states of participants and their perception of chocolate ice cream. Consuming ice cream while listening to liked music (LM) and while listening to the combination of liked music and pleasant sound (LMPS) conditions gave rise to more positive emotions than listening to just pleasant sound (PS). Consuming ice cream during the LM condition resulted in the longest duration of perceived sweetness. On the other hand, PS and LMPS conditions resulted in cocoa dominating for longer. Bitterness and roasted were dominant under the disliked music and unpleasant sound (DMUS) and DM conditions respectively. Positive emotions correlated well with the temporal sensory perception of sweetness and cocoa when consuming chocolate ice cream under the positively valenced auditory conditions. In contrast, negative emotions were associated with bitter and roasted tastes/flavours under the negatively valenced auditory conditions. The combination of pleasant music and non-musical sound conditions evoked more positive emotions than when either was presented in isolation. Taken together, the results of this study support the view that sensory attributes correlated well with emotions evoked when consuming ice cream under different auditory conditions varying in terms of their valence.

## 1. Introduction

Everyday eating and drinking experiences involve multiple interrelated sensory inputs associated with the food or drink itself [1]. What is more, a growing body of literature now recognises the importance of environmental sounds [2,3,4,5], and music [6,7,8,9] on food perception. Furthermore, emotions can be elicited by both music [6,7,8,9,10] and sounds [3,5,11,12]; though, to date, the influence of the two when combined has received little attention. Intriguingly, emotions have been shown to influence the sensory perception of both chocolate [13,14] and chocolate ice cream [7].

Environmental sounds can influence the flavour and pleasantness of food. For instance, Kantono et al. [2] examined the influence of different environmental sounds (café, fast food restaurant, and bar) on the pleasantness of chocolate ice-cream and reported that positively-valenced sounds significantly increased the pleasantness of the participants’ tasting experience. Lin et al. [3] further demonstrated that the valence evoked by listening to a pleasant park sound was positively correlated with the sweetness of the chocolate ice cream. Furthermore, the arousal associated with the sounds of a bar was positively correlated with bitterness, roasted, and cocoa attributes.

Food perception can also be influenced by soundtracks and music. For instance, Carvalho et al. [15] reported that a putatively ‘creamy’ soundtrack, which was more liked than a ‘rough’ soundtrack, increased ratings of chocolate liking, sweetness, and creaminess. Meanwhile, Wang and Spence [16] found that participants rated wine as tasting sweeter while listening to a putatively ‘sweet’ soundtrack as compared to a ‘sour’ soundtrack, although it should be noted that this crossmodal effect was only documented for the sweeter of the two wines tested. Kantono et al. [8] reported that disliked music significantly decreased the pleasantness of chocolate ice cream, while liked music increased pleasantness. Further studies have revealed that the consumption of chocolate ice cream under liked music conditions is associated with sweet taste [6,7,8]. On the other hand, the perception of bitter [6] and creamy [7,8] dominated when participants consumed ice cream under disliked music conditions. These studies confirmed that positive sensory attributes were perceived under liked music conditions. However, the influence of a mixture of musical and non-musical sounds on temporal perception has not been investigated.

A relatively independent literature has shown that how food is perceived is influenced by the emotional state of the consumer, even when consuming food under different environmental conditions. For instance, Thomson, Crocker, and Marketo [17] demonstrated that cocoa was associated with power and energy; bitter with confidence, adventure and masculinity; smoky/burnt with arrogance, seriousness, traditionalism, and pretentiousness; vanilla with sensuality, fun, and luxury; and creamy and sweet with fun, comfort, and easy-goingness. Jager et al. [14] found that plain dark chocolates were higher in cocoa, dry, bitter, and sour attributes, and elicited calm, bored, aggressive, and guilty emotions. Gunaratne et al. [18] further reported that sweet chocolate was associated with emotional terms such as “happy”, “joyful”, “pleasurable”, “affectionate”, “enjoyment”, and “comforting”, while bitter chocolate was associated with “natural”, “luxurious”, “relaxed”, “healthy”, and “satisfied”.

Human emotions are highly attuned to changes in the acoustic environment [19]. The affective dimensions of valence and arousal have been studied by many researchers using both natural and anthropic sounds [20]; environmental sounds [3,12]; natural, human sound, and traffic noise soundscapes [11]; and music [8,10] to document how they influence emotions. However, few studies have investigated how food consumed when listening to a mix of sounds influenced people’s emotions. Xu et al. [5] found that listening to a café sound mixed with forest or bird soundscapes while eating chocolate ice cream was associated with positive emotions such as amusement, happiness, enjoyment, love, and satisfaction. However, when mixed with an incongruent machine sound, negative emotions were evoked.

The role of music in evoking emotions has been studied extensively. Flores-Gutiérrez et al. [21] evaluated the subjective feelings elicited by different pieces of music and found that positive emotions were positively correlated with pleasant music conditions. Conversely, negative emotion was reported to be positively correlated with unpleasant music conditions. However, only in the last six years have studies attempted to determine how consumption under music conditions varies in valence-influenced emotions. Kantono et al. [6,7,8] measured emotional responses after the consumption of chocolate gelato while listening to music varying in likeability. Liked music evoked higher ratings of positive emotions and the disliked music condition evoked higher ratings of negative emotions after the consumption of chocolate ice cream.

### Objectives

The novelty of this study is to provide an understanding of how a mixture of musical and non-musical sounds varying in pleasantness can influence temporal changes in food perception. This is because in the real world, music is rarely heard without interference from other sounds. This study sought to document the influence of musical and non-musical sounds, and a mixture of the two, on the sequence of sensations associated with tasting chocolate ice cream. Additionally, the way in which both musical and non-musical sounds varying in terms of their valence (i.e., pleasantness), influenced the emotional states of participants was also investigated to understand how flavour perception was influenced. It was hypothesised that sounds and music with positive valence would evoke positive affective states and thus enhance the more desirable sensory attributes of food. On the other hand, unpleasant music and/or sound will likely induce negative affective states and, in turn, increase the undesirable sensory attributes of ice cream.

## 2. Materials and Method

### 2.1. Ice Cream Preparation and Presentation

The bittersweet chocolate ice cream samples were formulated using a mixture of cream (51.54%), milk (15.38%), sugar (15.38%), and cocoa powder (7.69%). These ingredients were poured into a Cuisinart ICE-100 Compressor Ice Cream and Gelato Maker (Conair, Connecticut Corporation, Stamford, CT, USA) and churned until the mixture thickened. The ice cream was transferred to covered foam cups and stored in a commercial-grade freezer (Fisher and Paykel, Auckland, New Zealand) at −18 °C for at least 24 h prior to testing to ensure the consistency of the sample. A scoop of ice cream (5.0 ± 0.8 g) was served individually in a 25 mL plastic portion cup. All samples were tempered for one minute at room temperature prior to serving. The serving temperature (−12 ± 2 °C) was strictly monitored to maintain consistency [22]. Sample presentation was randomised and counterbalanced across panellists [23].

### 2.2. Ethics Statement

Ethics approval by the Auckland University of Technology Ethics Committee (AUTEC 12/202) was obtained for this study. Participants were provided with informed consent forms prior to the start of the study. They were rewarded with supermarket vouchers for their participation in the experiment.

### 2.3. Participants

Sixty-four participants (28 males, 36 females) aged between 20 and 40 years old were recruited (mean age = 27.63 years; *SD* = 4.55 years). The number of participants was determined using a Cohen’s *d* of 0.8 giving a statistical power of 0.90–0.95 [24]. Participants were asked whether they had hearing difficulties in the consent form. All participants were consumers of chocolate ice cream, and none reported any medical conditions that could affect their taste perception.

### 2.4. Music and Sound Selection

#### 2.4.1. Music Selection

Panellists rated their liking of fourteen different music genres [25] as shown in Section A.1. A 45-s segment of each genre was played. Studies by Kantono et al. [6,7,8] found that listening to music for 45 s was the optimum time for the rating of the pleasantness of ice cream. Music was selected and categorised for each genre according to the Apple iTunes music classification system. A preliminary trial was carried out with sixty people (21 to 40 years old) who regularly listened to music to identify the song that best represented each genre (see Section A.1). The music samples were then modified using Adobe Audition CC version 11.1.1.3 (Adobe Inc., San Jose, CA, USA) to scale the average sound pressure level (SPL) to 70 dB SPL. Music was played through a Sennheiser headset (Series HD 518: Sennheiser Electronics GmbH and Co. KG, Wedemark, Germany), and participants rated their liking of each genre (see Section A.1) using a 100-mm unstructured line scale anchored with ‘extremely dislike’ and ‘extremely like’ at each end of the scale. The lowest and highest liking scores were classed as disliked (DM) and liked (LM) music respectively, and this approach accounted for individual tastes in music, with each individual listening to liked and disliked music as they consumed ice cream.

#### 2.4.2. Sound Selection

A preliminary study carried out with sixty-eight people (21 to 40 years old) agreed on fourteen sounds that best represented pleasant and unpleasant sounds (see Section A.2). The soundscapes used in this study were purchased from Soundsnap (https://www.soundsnap.com/, accessed on 11 April 2018). measured pleasantness in the study. Panellists rated the pleasantness of 14 different sounds using an unstructured line scale (0 mm = extremely unpleasant; 100 mm = extremely pleasant) according to Kantono et al. [9]. The pleasant (PS) and unpleasant (US) sounds identified by each panellist were then used in the main experiment. Subsequently, seven sounds were classified as pleasant (rated between 7.71 (*SD* = 1.46) and 9.13 (*SD* = 0.81)) and the remaining seven sounds as unpleasant (rated between 1.09 (*SD* = 0.82) and 2.62 (*SD* = 1.80)).

Each sound was played to each participant over a pair of Sennheiser Series HD 518 headphones (Sennheiser Electronics GmbH and Co. KG, Wademark, Germany) using a standard PC sound card during the experiment. The order in which the various auditory stimuli were delivered was randomised and counterbalanced to minimise any order effects. All soundscapes were high-pass filtered using Adobe Audition CC version 11.1.1.3 (Adobe Inc., San Jose, CA, USA) to balance sound pressure levels across all sound samples. The root mean square amplitudes of the audio samples were standardised to an internal reference in order to achieve equivalent average sound pressure levels across all audio samples, and later scaled to 70 dB of sound pressure level (SPL), using a Brüel and Kjær sound meter (Brüel and Kjær, Nærum, Denmark).

#### 2.4.3. Selection of Music and Sound Mix

The mixtures of liked music and pleasant sound (LMPS), and the mixtures of disliked music and unpleasant sound (DMUS) used in the experiment were unique to the individual panellists. The music and soundscape samples were compiled in audio format (using Adobe Audition CC version 11.1.1.3., Adobe Inc., San Jose, CA, USA). Each music file and sound file was imported into Audition CC, and then exported as a mixture of music and sound files. The sound file was then presented using a pair of Sennheiser Series HD 518 headphones (Sennheiser Electronics GmbH and Co. KG, Wademark, Germany) equipped with a standard PC sound card, with a sound intensity of 70 dB SPL. For the music and sound mixtures, the orders of presentation were counterbalanced.

### 2.5. Temporal Dominance of Sensation (TDS)

The effects of music and/or sound on the temporal aspects of multisensory flavour perception of chocolate ice cream was investigated using the temporal dominance of sensations (TDS) method [26]. This is a dynamic temporal sensory method designed to measure dominance of a pre-defined list of attributes at a given point of time. “Dominance” is defined as the sensation that gives the “most striking perception at a given time” [27]. This method has been widely used with consumers because it requires less training [28].

Sensory attributes were presented in combination with an unstructured line scale with anchors at each end labelled with ‘none’ and ‘intense’. The attributes being measured in the study were sweetness, bitterness, cocoa, creamy, milky, roasted, and vanilla. The description and reference standards for these attributes are summarised in Table 1. Each TDS session lasted up to 45 s, which coincided with the participants listening to the different sound conditions. A mandatory 45-s silent break was given between each auditory condition the participants were exposed to.

#### Panel Familiarisation

Panel familiarisation was carried out according to Hutchings et al. [29] over three sessions that lasted 8 h. Familiarisation sessions were carried out over two days in the same week. On the first day, the panellists spent three hours each for the first and second familiarisation sessions. On the second day, panellists were familiarised with the TDS method for a further two hours in the third session. Panellists were informed that they would be listening to music and sounds while consuming chocolate ice cream.

In the first session, the panellists familiarised themselves with the measurement of flavour sensations using the TDS procedure and were introduced to the concept and measurement of dominance. The ‘dominance’ of attributes was defined as that attribute capturing their attention at a given moment in time, and panellists were informed that dominance might change when a new sensation became more salient [26,30]. In addition, panellists rated the intensity of the selected dominant attribute using an unstructured line scale, anchored with “none” and “extreme” at each end. In the second session, panellists were familiarised with the sensory attributes of the chocolate ice cream using a list of definitions and reference standards for each attribute (Table 1). In addition, panellists were familiarised with emotion attributes (shown in Section A.3) that they selected after carrying out TDS. The third session focused on familiarising panellists with the TDS technique. Panellists evaluated changes in sensory perception of chocolate ice cream using TDS under different sound conditions. This allowed them to familiarise themselves with the computer interface and the TDS methodology.

### 2.6. Emotional Responses

In this study, the emotions that fall within the valence–arousal–dominance (VAD) model introduced by Russell and Mehrabian [31] were used. The authors stated that any emotions in this model can be represented in terms of three fundamental dimensions: valence, which expresses a pleasant or unpleasant feeling about something; arousal, which describes the level of affective activation; and dominance, which reflects the level of control of the emotional state. In addition, CATA questions were used to obtain verbal self-report measures (i.e., emotion words) to evaluate consumers’ emotional responses. This method has investigated consumers’ emotional responses to commercial blackcurrant squashes [32]. The authors found that the consumer-defined lexicon provided a rich and more balanced list of positive and negative emotions specific to the products and yielded emotional data that clearly discriminated between them.

#### 2.6.1. Affective Responses

The participants were presented with an unstructured line scale (40 cm) with a continuous line scale for the attributes of valence, arousal, and dominance [8], and these measurements were used to determine the panellists’ affective responses towards the sound stimuli when consuming ice cream [2]. Valence was described as the pleasantness—anchored from unpleasant to pleasant—of the sample. Arousal was described as the intensity of emotion evoked by the sample, anchored from calm to excited; and dominance was described as the extent to which the sample grabs one’s attention, anchored from controlling to not controlling attention [33].

#### 2.6.2. Measurement of Emotions

A preliminary study was carried out using a focus group consisting of 36 participants (16 males, 20 females) between 21 and 34 years of age to capture the emotions experienced during consumption of chocolate ice cream under the different sound conditions. In this study, emotional terms without reference to food or sound were selected from the profile of mood states [34], multiple affect adjective checklist (revised) [35], positive and negative affect scale [36], and the Geneva affect label coder [37]. Out of an initial 150 emotion terms, the twenty most relevant and well-understood terms, as selected by panellists, were retained.

The check-all-that-apply (CATA) approach consisted of a list of 20 terms, all comprising emotional attributes (see Section A.3). There were eleven positive emotional terms (active, at ease, calm, energetic, enthusiastic, excited, interested, joy, pleasant, relaxed, and satisfaction), and nine negative emotional terms (annoyed, anxious, boredom, lonely, restless, tired, unable to concentrate, uneasy, and unhappy). Consumers were instructed to check all the terms from the list that they considered appropriate to characterise their emotional responses during consumption of chocolate ice cream under different music and sound conditions. The order in which the CATA terms were presented was balanced between and within participants, following a Williams’ Latin square experimental design. The participants were asked to check all terms they considered appropriate to describe their emotions after consuming each ice cream sample under different sound conditions using the Fizz software (Biosystèmes, Couternon, France) for sensory analysis.

### 2.7. Experimental Procedure

The measures in this study were obtained at three different stages during a single experimental session that typically lasted sixty minutes (Figure 1). First, participants made subjective ratings of the 14 music genres and 14 sounds. Participants were instructed to play each music file and listened to the whole piece of music before rating their liking of each music genre. After a 5-min break, participants were instructed to play each sound file and listened to the whole piece of sound before rating the pleasantness of each sound. Participants then had a 10-min break before proceeding with the next task.

The next task involved TDS evaluation of the ice cream under different sound conditions. A single TDS trial lasted for 45 s to measure the flavour dynamics of the ice cream. On-screen instructions were provided in order to minimise eating variations. Once participants were ready, they clicked on the unstructured line scale for the dominant attribute to start rating the samples. On-screen instructions were provided on how to consume the chocolate ice cream samples. Pertinently, the music and/or sound was automatically played when participants first clicked the TDS button on screen. In the third stage during a forced 5-min break between samples, participants subjectively rated their affective responses after each TDS trial using an intensity scale, and selected emotions using a checklist.

### 2.8. Data Analysis

#### 2.8.1. Temporal Dominance of Sensation (TDS) Curves

In this study, TDS dominance curves summarised the dominance ratings of all sensory attributes over time using an in-built spline-based smoothing algorithm resident in the FIZZ software (Fizz 2.61.1, Biosystemes, Bourgogne, France). The curves displayed the percentage of panellists who recognised the prescribed attributes as being dominant at a given time [27].

The TDS time period is presented as standardised time (ST), and data are converted to percentages (0–100%; [38]). Time data was standardised to a score between 0 and 100 for each participant, where 0 represents the first click on the line scale, and 100 when the panellist either clicked stop or when the recording stopped automatically. The calculations of significant and chance levels were carried out according to Pineau et al. [27].

Chance level is the dominance rate that an attribute can obtain by chance. The value of chance level (*P*_0_) is equal to 1/p, p being the number of attributes, which in this case was seven. Therefore, the value of chance level is 14.29%. Significance level is the minimum value this proportion should be equal to be considered as significantly higher than *P*_0_. It is calculated using the confidence interval of a binomial proportion based on a normal approximation. The formula is shown below:$$P_{s\;}=P_{0}+1.645\;\sqrt{\frac{P_{0}\;\left(1-P_{0}\right)}{n}}$$
where *Ps* is the lowest significant proportion value (α = 0.05) at any point in time for a TDS curve, *n* is the number of subject times replication (64). Therefore, the value of significance level was calculated to be 21.48%.

#### 2.8.2. Measure of Affective States (Valence, Arousal, and Dominance; VAD)

A one-way ANOVA was performed on the valence and arousal measures as a function of the different auditory conditions. Post hoc comparison using Tukey’s (HSD) test was applied when omnibus statistical significance was observed (i.e., *p* < 0.05).

#### 2.8.3. Multidimensional Alignment (MDA)

Multidimensional alignment (MDA) was used to evaluate CATA findings [39,40] by determining the cosine values between the auditory conditions and emotion attributes originating in the correspondence analysis of the CATA questions. By calculating the cosine of the angle (ranging from −1 to 1) formed between each attribute and condition, it was possible to determine attributes that have a strong relationship with each sound condition. Absolute cosines below 0.707 indicate hardly any relationship at all [39].

#### 2.8.4. Canonical Variate Analysis (CVA) of Sensory and Emotion Data

CVA was performed on sensory flavour perception and emotional data. A CVA, which minimises residual variability and maximises the distances between samples [41], was carried out using XLSTAT (version 2019.1.2 (Addinsoft Inc., New York, NY, USA). Additionally, multivariate analysis of variance (MANOVA) tests were used to determine if significant differences exist between the seven different music and/or sound conditions in terms of standardised durations of flavour.

#### 2.8.5. Partial Least Square Path Modelling (PLS-PM)

PLS-PM was performed using the XLSTAT PLSPM module Version 2019 (Addinsoft Inc., New York, NY, USA) to construct cause-and-effect models. PLS-PM summarised the relationships between emotional responses, affective states, and sensory measurements in this study. To assess whether the model was reliable, two goodness of fit (GoF) statistics, Cronbach’s α and Dillon-Goldstein’s *rho*, were obtained for each manifest variable according to Henseler and Sarstedt [42]. GoF is calculated as the geometric mean of the average communality and the average *R*^2^ value. Cronbach’s alpha is a coefficient to measure how well a related block of indicators measure their corresponding latent construct. Dillon-Goldstein’s *rho* focuses on the variance of the sum of variables in the block of interest. Additionally, the *R*^2^ value of the latent variable was also calculated. A value greater than *R*^2^ = 0.7 for all three statistics indicates sufficient reliability [43].

## 3. Results

### 3.1. Affective Dimensions

#### Changes in Affective Dimensions with Music and Sounds Stimulus

As shown in Figure 2, differences were observed in terms of the affective dimensions of valence and arousal associated with music and/or sound conditions. Specifically, significant differences in terms of valence (*F*_(6,447)_ = 70.38, *p* < 0.001, partial eta squared = 0.49), and arousal (*F*_(6,447)_ = 39.29, *p* < 0.001, partial eta squared = 0.35). The silent, LM, PS, and LMPS conditions were significantly higher in valence compared to DM, US, and DMUS conditions. In terms of arousal, the DM, US, and DMUS conditions were significantly higher in arousal than silent, LM, PS, and LMPS conditions.

### 3.2. Effect of Music and Sound Conditions on Emotions Evoked

MDA was applied to determine the cosine values between −1 and +1 [40] to show how sound conditions influenced emotion attributes when consuming chocolate ice cream. Table 2 highlights the emotions that are positively and negatively correlated with the combined auditory conditions using the first two dimensions of the CA bi-dimensional map. The relationship between emotional attributes and auditory conditions with cosine values > 0.707 are further explained [39]. This is because the authors reported that for interpretation, absolute cosines below 0.707 (=cos(45°) = −cos(135°)) indicate hardly any relationship at all.

The silent, LM, PS, and LMPS conditions were positively correlated with valence—pleasant (+), joy (+), and relaxed (+). Silent, LM, and LMPS conditions were positively correlated with interested (+), active (+), and satisfied (N), and PS had a strong positive correlation with calm (+) and at ease (+). In contrast, DM, US, and DMUS conditions were positively correlated with arousal—unable to concentrate (−), annoyed (−), restless (−), tired (−), uneasy (−), unhappy (−), anxious (−), boredom (−), and lonely (−).

### 3.3. TDS Result

Figure 3 shows the spline-smoothed TDS dominance curves describing the dominance rate of various chocolate ice cream attributes consumed in the silent and other auditory conditions. The calculated chance and significant levels were between 14.29% and 21.48%, respectively. Changes in attributes above 21.48% (the significance level) were explained and discussed. Sweetness was the first dominant attribute under all control sound and music conditions, except for the DMUS condition. However, changes in sweetness dominance were observed during consumption.

#### 3.3.1. Silent Condition (Control)

For the silent condition, a higher duration of sweetness was evident, with a maximum dominance rate of 76.60% at the start consumption that became less dominant until 30% standardised time (ST). Cocoa increased between 26–52% ST and 55–100% ST, reaching a maximum dominance rate of 45.30% at 86% ST.

#### 3.3.2. LM Condition

In the LM condition, a longer duration of sweetness domination was noted, with a maximum dominance rate of 71.90% at the start consumption that decreased until 61% ST. Cocoa increased significantly around mid-consumption (62–100% ST), reaching a maximum dominance rate of 46.90% at the end of consumption.

#### 3.3.3. DM Condition

When listening in the DM condition, sweetness was perceived as being dominant only at the start of consumption, with a maximum dominance rate of 50.00% at 0% ST that decreased until 3% ST. Bitterness was the next dominant attribute from 3% ST until 41% ST, reaching a maximum dominance rate of 48.40%. Roasted was significantly dominant between 42–78% ST, reaching a maximum dominance rate of 35.9% at 64% ST. Finally, bitterness was dominant from 79–100% ST, reaching a maximum dominance rate of 29.70% at 100% ST.

#### 3.3.4. PS Condition

In the PS condition, sweetness was the first dominant attribute, starting at a maximum dominance rate of 75% and slowly decreasing from 0–34% ST. Cocoa was significantly dominant next, from 35–100% ST, and ranging between 26.60–39.10%.

#### 3.3.5. US Condition

During the US condition, sweetness was dominant at the start and decreased from a maximum dominance rate of 64.10% between 0% and 10% ST. Bitterness was the dominant attribute thereafter, dominating between 11–27% ST and 39–55% ST, reaching a maximum dominance rate of 40.60% at 44% ST. Roasted was the dominant attribute at several points during the consumption period: from 27–38% ST, and 56–100% ST, reaching a maximum rate of 32.80% at 35% ST, and 43.80% at 94% ST.

#### 3.3.6. LMPS Condition

In the LMPS condition, sweetness was the first dominant attribute that started at a maximum dominance rate of 75% that slowly decreased from 0–45% ST. Cocoa was the next dominant attribute from 62% ST to 100% ST, reaching a maximum rate of 45.30% at 88% ST.

#### 3.3.7. DMUS Condition

In the DMUS condition, bitterness was dominant from 0% to 36% ST, reaching a maximum of dominance rate of 56.30% at 7% ST. Next, roasted became the dominant attribute from 37–58% ST, reaching a maximum rate of 31.30% at 52%ST. Finally, bitterness was dominant from 59–100% of ST, reaching a maximum dominance rate of 39.10% at 73% ST.

### 3.4. Canonical Variate Analysis (CVA) of Sensory and Emotional Responses to Chocolate Ice Cream Coinciding with Different Music and Sound Conditions

CVA summarised the standardised duration of flavour perception when chocolate ice cream was consumed under different music and sound conditions. Figure 4 shows the first two canonical variates, which explained 79.40% of the data. The Hotelling–Lawley MANOVA analysis results revealed significant differences between the standardised duration of flavour perception of chocolate ice cream in the silent and the six music and sound conditions (*F*_(180,1992)_ = 5.07; *p* < 0.0001, partial eta squared = 0.31).

Figure 4 shows a clear separation of ice cream consumed under different music and sound conditions in terms of emotions and sensory perception, with the first factor explaining 60.09% of the variance in the data. The LM and LMPS conditions were correlated to the affective states of valence and associated with the positive emotions of pleasant, relaxed, satisfied, joy, interested, and active. The silent and PS conditions were associated with positive emotions of being calm, at ease, and pleasant on the other. These conditions were associated with sweetness, cocoa, and milky flavours. DM, US, and DMUS conditions were correlated with the affective states of arousal and to the negative emotions of uneasy, unhappy, anxious, tired, lonely, boredom, restless, annoyed, and unable to concentrate. These conditions were associated with bitterness and roasted flavours. The second factor (F2) accounted for 19.31% of the variance and further differentiated between PS and LM conditions. LM was associated with positive emotions of enthusiastic, energetic, active, and excited. PS were associated with fewer emotions (pleasant, at ease, and calm).

### 3.5. PLS-PM Analysis of Affective States, Emotion Measure, and Sensory Perception

PLS-PM was used in this study to illustrate the emotional mechanism on how affective states and emotional responses can influence sensory perception as shown in Figure 5. PLS-PM is mainly used to construct cause-and-effect models. The use of PLS-PM emerged as an important tool to explain how the merging of food choice questionnaire and sensory perception influenced consumer behaviour of low sugar products [44], to demonstrate the relationships between physiological measurements, self-reported emotions, and sensory measurements when consuming chocolate gelato under different music conditions [7], and for linking data about consumers, products and acceptance [45]. One of the main advantages of PLS-PM is that it calculates reliability and validity at the same time [46]. Cronbach’s alpha, Dillon-Goldstein’s *rho*, and *R*^2^ values for each variable were greater than 0.7, indicating overall reliability of the measured variables [43]. The relative GoF measured value (GoF = 0.801) was similar to the GoF of the bootstrapping model (GoF_®_ = 0.812). These high values reflect a good fit of the model to the data.

Valence was significantly positively correlated with positive emotions (pleasant, in-terested, joy, relaxed, active, at ease, calm, satisfied, excited, enthusiast, and energetic). Positive emotions were in turn positively correlated with sweetness, while showing nega-tive correlations with bitterness and creaminess. Arousal, on the other hand, had signifi-cant positive correlations with negative emotions (annoyed, restless, tired, uneasy, un-happy, anxious, boredom, lonely, and unable to concentrate). Negative emotions were in turn positively correlated with bitterness and roasted, while being negatively correlated to sweetness, milky, creaminess, and cocoa.

## 4. Discussion

### 4.1. LM, PS, and LMPS Were Positive Consumption Conditions

The Silent, LM, PS, and LMPS conditions had significantly higher valence ratings (i.e., were more pleasant) as compared to DM, US, and DMUS conditions. Kantono et al. [8] also reported that liked music was rated as being significantly more pleasant as compared to disliked and neutral music conditions when consuming chocolate gelato. Meanwhile, Lin et al. [3] reported that listening to a pleasant park soundscape when consuming chocolate ice cream had significantly high valence. Vitale et al. [12] demonstrated that pleasant environmental sounds were associated with higher median ratings of pleasantness. Similarly, in the present study, nature sounds (waterfall, birds, thunder, rain, and burning fire sounds) were typically rated higher in pleasantness by the panellists than traffic noise. This is consistent with another study reporting that a natural soundscape was associated with higher pleasantness ratings compared to human sound and traffic noise soundscapes [11].

### 4.2. DM, US, and DMUS Conditions Were High in Arousal after Consuming Chocolate Ice Cream

DM, US, and DMUS conditions resulted in significantly higher arousal ratings compared with the silent, LM, PS, and LMPS conditions. This finding is consistent with Kantono et al. [8], who reported that disliked music was significantly more arousing than neutral and liked music conditions when consuming chocolate gelato. Lin et al. [3] further reported that an unpleasant bar soundscape scored significantly higher in arousal compared to park, café, fast food restaurant, and food court sound conditions.

### 4.3. LM and LMPS Evoked More Positive Emotions than PS

One interesting finding to emerge from this research is that the LM and LMPS conditions evoked more positive emotions than PS. Using the Cochran’s Q test (refer to Section A.4), there was no significant differences between LM and LMPS in terms of positive emotions. However, LM evoked significantly higher positive emotions than PS. Music is an important part of many people’s lives [47], due to its power to evoke strong emotions and influence moods [48]. During the presentation of unpleasant music, the amygdala, hippocampus, parahippocampal gyri, and temporal poles are activated [49], all of which are central to sympathetic response circuits associated with negative emotions [50]. Music is often played in the home, where people have control over what is heard, and is associated with the safe environment that typifies most homes. In contrast, sound often signals some-or-other environmental contingency, and, as such, attended sounds are typically described as being ‘eventful’. Thus, while the focus upon music is on the characteristics of the music itself (e.g., beat, melody, etc.), everyday sound is not so much used for aesthetic purposes, as to help provide the listener with a sense of what is going on in their immediate environment [51]. Gray [52] argues that environments free from stressors, such as the home, typically induce positive emotions and an approach response known as the behavioural activation system. This system is sensitive to classical conditioning, whereby a stimulus such as music may become associated with the positive affect experienced in safe and stress-free environments. As such, we speculate that our findings reflect a general preference of participants for music, which may have greater hedonic/aesthetic value, than everyday sounds that act more as alerting/informational/warning signals.

### 4.4. Music and Sound Conditions Differing in Valence Influence the Dominance of Sensory Sensations When Consuming Chocolate Ice Cream

The results of this study (see Figure 4) demonstrate that the sensory-perceptual experiences of chocolate ice cream were affected by the different music and sound conditions. Consuming chocolate ice cream under LM, PS, and LMPS conditions resulted in the prolonged dominance of sweetness and cocoa attributes. In addition, the duration of sweetness was shorter, and cocoa was higher under the PS condition as compared to the LM and LMPS conditions. These results agree with those of Kantono et al. [6,9] who reported that milk, dark chocolate, and bittersweet chocolate gelati were significantly dominated by sweetness in the liked music conditions. Kantono et al. [7] further noted a longer duration of sweetness when consuming chocolate ice cream when listening to liked music. Kantono et al. [8] also reported cocoa and sweetness were the most cited when eating chocolate ice cream in the natural eating environment, and less cited in both immersive and laboratory environments. Note that our findings are consistent with the studies of Carvalho et al. [15], Wang and Spence [16], and Lin et al. [3].

Another important finding to emerge from the present study is that consuming chocolate ice cream under DM, US, and DMUS conditions resulted in a longer dominance of bitterness and roasted attributes. Similarly, Kantono et al. [6,7] reported that listening to disliked music was associated with a higher dominance rate of bitterness in dark chocolate, bittersweet, and milk chocolate gelati. Kantono et al. [8] further found that bitterness was most cited in the laboratory environment when listening to disliked music. Lin et al. [3] showed that roasted and bitterness were most cited under the least pleasant sound conditions (bar, fast food, and food court sounds) when consuming chocolate ice cream.

### 4.5. Relationship between Affective Response, Emotion and Sensory Perception

The relationship between affective responses, emotion ratings, and sensory perceptions were explored using PLS-PM. Valence was correlated with positive emotions that, in turn, were further correlated with sweetness. Similarly, Kantono et al. [7] reported that positive emotions of amusement, enjoyment, love, happiness, and satisfaction showed a significant positive correlation with the sweetness and milkiness of chocolate ice cream. Meanwhile, Thomson et al. [17] previously reported the link between sensory characteristics and emotions while consuming dark chocolate. The latter researchers noted that sweet was associated with fun, comforting, and easy-going emotions. Jager et al. [14] documented the temporal dynamics of sensory and emotional attributes during chocolate tasting. They found that the consumption of orange- and blueberry-flavoured chocolate samples resulted in the perception of sweet, fruity, and crunchy, and evoked positive emotions of being interested, happy, and loving. Gunaratne et al. [18] further demonstrated that sweet chocolate was associated with emotions such as happy, joyful, pleasurable, affectionate, enjoyment, and comforting. Therefore, it can be concluded that those chocolate products perceived with positive sensory attributes are associated with positive emotions.

Arousal was positively correlated with negative emotions that were further correlated with bitter and roasted tastes. Kantono et al. [8] also reported that the negative emotions of anger, contempt, disappointment, and disgust were correlated with bitterness when consuming chocolate ice cream. Jager et al. [14] reported that the two plain chocolates (70% and 85% cocoa) were perceived as tasting more bitter and evoked bored, aggressive, and guilty emotions. Gunaratne et al. [13] further demonstrated that bitter chocolate was associated with emotional terms such as natural, luxurious, relaxed, healthy, and satisfied.

## 5. Conclusions

This study set out to investigate how sounds and/or music varying in valence influenced both sensory attributes and emotional experiences evoked by chocolate ice cream. The positive valence (pleasant) LM and LMPS conditions evoked more positive emotions than the PS condition. More negative emotions were evoked in the negative valence (unpleasant) DM, US, and DMUS conditions. Consuming ice cream while listening to the highly positively valenced musical and non-musical sound conditions resulted in varying dominance and duration of sweet and cocoa that were correlated with positive emotions. On the other hand, negatively valenced (unpleasant) music and sound conditions resulted in the dominance of bitter and roasted attributes that were correlated with negative emotions. These findings enhance our understanding of how the flavours of food can be modulated using different music and sound conditions. The scope of this study was limited in terms of the type of food used; hence, it would be interesting to assess the effects of auditory cues on different foods. Further studies are needed to document how music and sound in real consumption environments may influence flavour.

## Figures and Tables

**Figure 1 foods-11-01784-f001:**
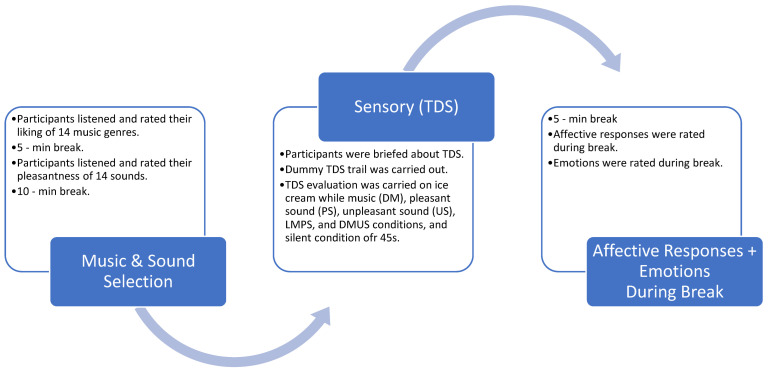
Experimental protocol for the study.

**Figure 2 foods-11-01784-f002:**
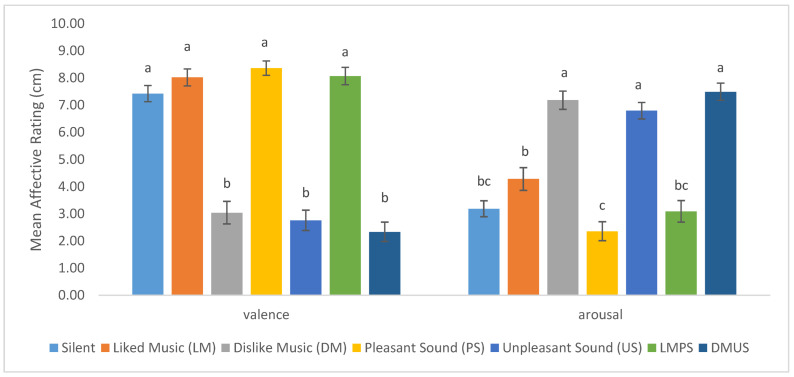
Mean ratings of valence and arousal during consumption of chocolate ice cream under various music and sound conditions. Different letters ^a^, ^b^, ^c^ indicates changes in mean affective ratings readings with different superscripts indicating significance (*p* < 0.05). Error bar indicates standard error.

**Figure 3 foods-11-01784-f003:**
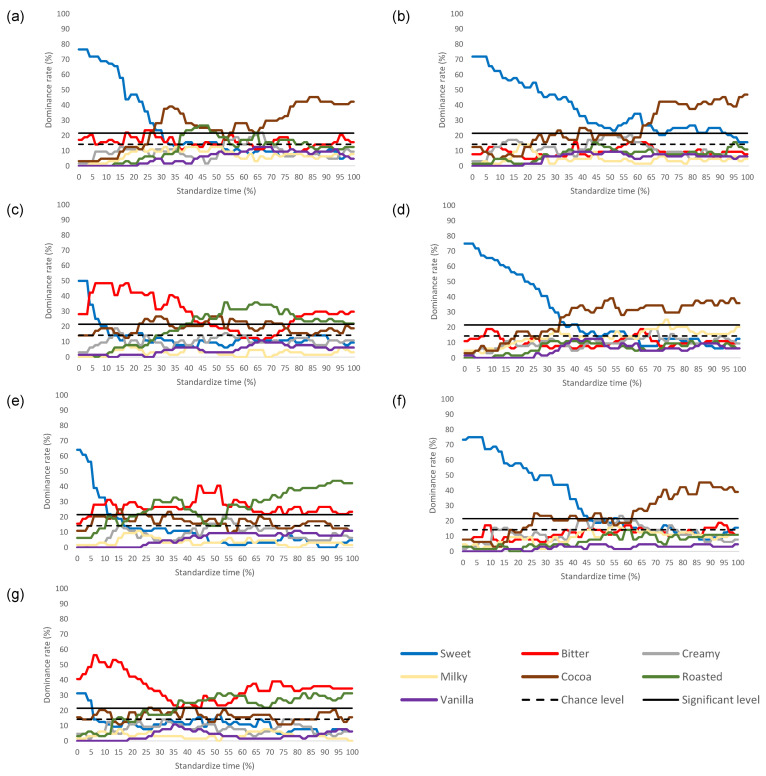
TDS dominance curves for chocolate ice cream consumed under: (**a**) silence; (**b**) liked music (LM); (**c**) disliked music (DM); (**d**) pleasant sound (PS); (**e**) unpleasant sound (US); (**f**) LMPS; (**g**) DMUS conditions. Panel dominance rates (%) of the seven sensory attributes presented in the TDS sessions are expressed as the % of standardised time. According to Pineau et al. [27], the curves represent the average dominance rates for all the panels (N = 64). The calculated chance and significance levels were between 14.29% and 21.48%, respectively.

**Figure 4 foods-11-01784-f004:**
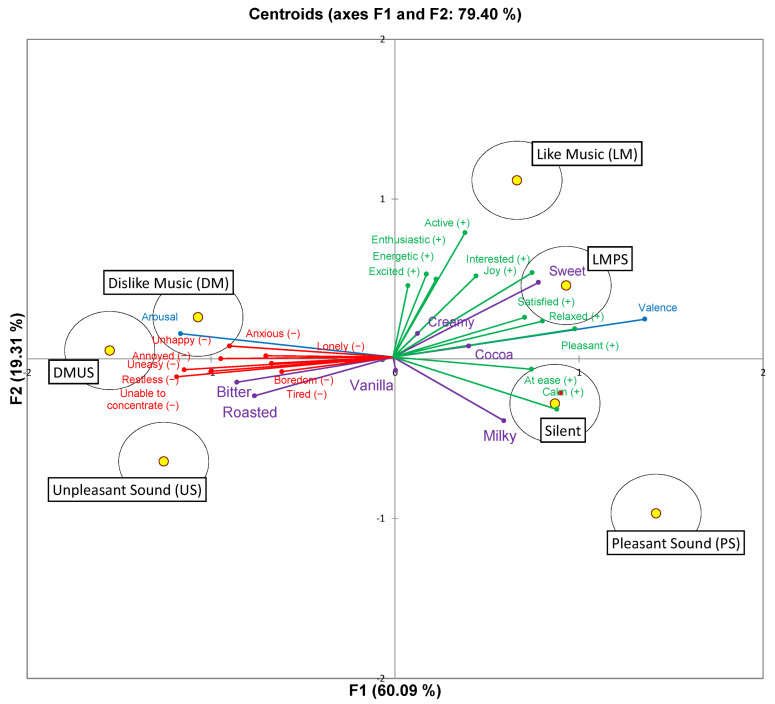
Joint canonical variate analysis (CVA) scores and loadings plots of dominance durations of sensations and emotional responses. A Hotelling–Lawley MANOVA test showed significant sample differences (*F*_(180,1992)_ = 5.07, *p* < 0.01, partial eta squared = 0.31) based on sensory attributes and emotional responses. To aid visualisation, positive emotional responses are labelled green, negative emotional responses are labelled in red, affective measures are labelled in light blue, and sensory attributes are labelled in orange.

**Figure 5 foods-11-01784-f005:**
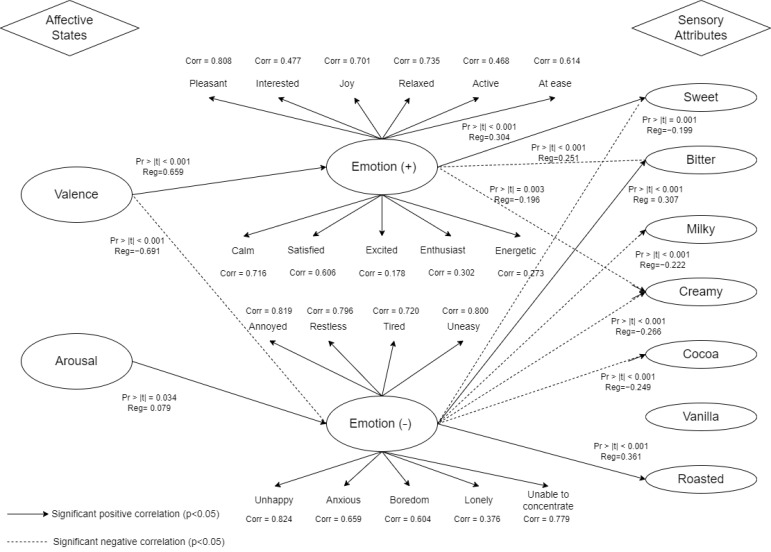
PLS-PM showing only the significant correlations between affective responses, self-rated emotion measures, sensory attributes, and perceptual correlates. To aid visualisation, positive correlations are indicated with continuous lines, and negative correlations with dashed lines.

**Table 1 foods-11-01784-t001:** Sensory attributes and descriptions of chocolate ice cream.

Sensory Attributes	Description	Reference Standard
Sweet (taste)	Taste associated with sugar	Hershey milk chocolate
Bitter (taste)	Taste associated with caffeine or quinine solutions	Hershey dark chocolate
Cocoa (flavour)	Characteristic flavour associated with cocoa	Hershey milk chocolate
Milky (flavour)	Characteristic flavour associated with milk	Fresh milk (Anchor™, New Zealand)
Creamy (texture)	Texture associated with cream	Fresh cream (Anchor™, New Zealand)
Vanilla (flavour)	A woody, slightly chemical aroma associated with vanilla bean	Heilala Vanilla pure vanilla extract + fresh milk (Anchor™, New Zealand)
Roasted (flavour)	A burnt, somewhat bitter character present in a product that has been cooked at a high temperature, typical of very strong dark coffee	Hershey milk chocolate + three highly roasted coffee beans

**Table 2 foods-11-01784-t002:** Cosine values between music and sound conditions and the attributes used to describe the emotions, obtained by correspondence analysis (CA) for seven conditions using CATA terms. Values in green and orange indicate high positive and negative correlations respectively between the emotion attributes and ice cream samples consumed under different sound conditions.

	Silent	LM	DM	PS	US	LMPS	DMUS
Valence	0.891	0.769	−0.983	0.885	−0.883	0.919	−0.960
Arousal	−0.804	−0.788	0.912	−0.821	0.953	−0.951	0.897
Pleasant (+)	0.832	0.785	−0.972	0.908	−0.880	0.936	−0.973
Interested (+)	0.921	0.927	−0.727	0.364	−0.835	0.819	−0.835
Joy (+)	0.806	0.918	−0.867	0.733	−0.970	0.979	−0.949
Relaxed (+)	0.862	0.804	−0.976	0.885	−0.932	0.961	−0.964
Active (+)	0.734	0.913	−0.541	0.145	−0.751	0.731	−0.686
Calm (+)	0.591	0.507	−0.844	0.998	−0.732	0.768	−0.806
Satisfied (N)	0.998	0.782	−0.853	0.612	−0.857	0.833	−0.878
Enthusiastic (+)	0.658	0.934	−0.505	0.170	−0.743	0.698	−0.686
Energetic (+)	0.586	0.880	−0.444	0.042	−0.618	0.606	−0.588
At ease (+)	0.639	0.594	−0.907	0.986	−0.767	0.826	−0.852
Unable to concentrate (−)	−0.883	−0.871	0.929	−0.812	0.985	−0.961	0.961
Annoyed (−)	−0.904	−0.890	0.912	−0.787	0.986	−0.961	0.975
Restless (−)	−0.908	−0.896	0.929	−0.791	0.981	−0.969	0.981
Tired (−)	−0.872	−0.914	0.950	−0.776	0.911	−0.960	0.991
Uneasy (−)	−0.914	−0.888	0.952	−0.805	0.941	−0.961	0.997
Unhappy (−)	−0.893	−0.829	0.981	−0.773	0.844	−0.944	0.960
Anxious (−)	−0.884	−0.875	0.898	−0.775	0.894	−0.919	0.988
Boredom (−)	−0.781	−0.897	0.926	−0.752	0.882	−0.981	0.958
Lonely (−)	−0.827	−0.966	0.859	−0.677	0.918	−0.942	0.968
Excited (+)	0.428	0.828	−0.346	0.166	−0.618	0.580	−0.599

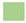
 Green values indicate high positive correlation (>0.707) of the emotion attribute with the respective sample. 
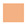
 Orange values indicate high negative correlation (<−0.707) of the emotion attribute with the respective sample.

## Data Availability

The data presented in this study are available on request from the corresponding author.

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
