# Peer review of "Musical and Non-Musical Sounds Influence the Flavour Perception of Chocolate Ice Cream and Emotional Responses"

_foods, 2022, doi:10.3390/foods11121784_

Round 1
Reviewer 1 Report
Abstract
- 23-24: “The combination of positive music and non-musical PLEASANT sound conditions evoked more positive emotions than when either was 24 presented in isolation”
Introduction
l.97: “ can influence changes in food PERCEPTION”
- 98-99: incomplete/incorrect sentence
l.105: “to understand HOW flavour perception was influenced.”
Materials and Methods
- 132: “ passed the sight and hearing check” : describe this check
- 193: explain the TDS method in more detail. Explain abbreviation. Not clear how TDS was performed, whether it involves a continuous or intermittent measurement for all scales at the same time?
- 2-05: “Sensory attributes were presented using an” , should this be “Sensory attributes were presented in combination with an” ?
Panel training: did you test whether the participants became reliable and reached stability, or could there still be a learning effect?
2.7.2. Measurement of Emotions: why create a new scale and not use a validated scale with food-related emotions like the EsSense25 ?
Results
Figure 2: Caption: “Differences in valence and arousal during..” should be “Mean valence and arousal ratings during” . What doe the letters a,b, c refer to?
Figure 3: are these data for N=64? Do the curves represent the mean of all individual dominance curves of all participants?
Discussion
Section 4.3: is there a significant difference between LM and LMPS, or does LM simply dominate PS?
Appendix 3:
Waterfall & jungle sounds is no longer available
Burning fire does not play
Since sites like this one are so volatile it would be better if the authors could provide the set of stimuli for download as additional material to guarantee their availability.
Author Response
Reviewer 1
Abstract
Q1. 23-24: “The combination of positive music and non-musical PLEASANT sound conditions evoked more positive emotions than when either was presented in isolation”
Responses:
The word “pleasant” has been added. Please refer to page 1, line 23 - 25.
Introduction
Q2. l.97: “ can influence changes in food PERCEPTION”
Responses:
The word “perception” has been added. Please refer to page 2 - 3, line 94 - 95
Q3. 98-99: incomplete/incorrect sentence
Responses:
The sentence has been corrected. Please refer to page 2 - 3, line 93 - 95
Q4. l.105: “to understand HOW flavour perception was influenced.”
Responses:
The word “how” has been added. Please refer to page 3, line 100 – 101.
Materials and Methods
Q5. 132: “ passed the sight and hearing check” : describe this check
Responses:
We did not carry out sight and hearing checks. People were just asked if they had difficulties with hearing in the consent form.
Please refer to page 3, line 127 - 128. Participants were asked whether they had hearing difficulties in the consent form.
Q6. 193: explain the TDS method in more detail. Explain abbreviation. Not clear how TDS was performed, whether it involves a continuous or intermittent measurement for all scales at the same time?
Responses:
We moved the “Experimental procedure” section from section 2.5 to section 2.7. Section 2.5 now explains the TDS procedure in detail.
Please refer to page 5, line 185 - 191.
The effects of music and/or sound on the temporal aspects of multisensory flavour perception of chocolate ice cream have been investigated using the temporal dominance of sensations (TDS) method. This is a dynamic temporal sensory method designed to measure dominance of a pre-defined list of attributes at a given point of time. "Dominance" is defined as the sensation that gives the “most striking perception at a given time” (Pineau et al, 2009). This method has been widely used with consumers because it requires less training (Galmarini et al, 2021).
Q7. 2-05: “Sensory attributes were presented using an” , should this be “Sensory attributes were presented in combination with an” ?
Responses:
The sentence has been changed to “Sensory attributes were presented in combination with an”. Please refer to page 5, line 192 - 193.
Q8. Panel training: did you test whether the participants became reliable and reached stability, or could there still be a learning effect?
Responses:
We did not train panels and assess reproducibility and reliability. Instead, we familiarised the panels according to Hutchings et al. (2017). We have rewritten this section to make it clearer.
Please refer to page 5, line 199 - 220.
Panel familiarisation was carried out according to Hutchings et al. (2017) over three sessions that lasted 8 hours. Familiarisation sessions were carried out over two days in the same week. On the first day, the panellists spent three hours each for the first and second familiarisation sessions. On the second day, panellists were familiarised with the TDS method for a further two hours in the third session. Panellists were informed that they would be listening to music and sounds while consuming chocolate ice cream.
In the first session, the panellists familiarised themselves with the measurement of flavour sensations using the TDS procedure and were introduced to the concept and measurement of dominance. The ‘dominance’ of attributes was defined as that attribute capturing their attention at a given moment in time, and panellists were informed that dominance might change when a new sensation became more salient (Labbe et al., 2009; Pineau et al., 2009). In addition, panellists rated the intensity of the selected dominant at-tribute using an unstructured line scale, anchored with “none” and “extreme” at each end. In the second session, panellists were familiarised with the sensory attributes of the by providing them a list of definitions and reference standards for each attribute. The definition and references for each attribute is summarised in Table 1. In addition, panellists were familiarised with emotion attributes shown in Appendix 3 that they selected after carrying out TDS. The third session focused on familiarising panellists with the TDS technique. Panellists evaluated changes in sensory perception of chocolate ice cream using TDS under different sound conditions. This allowed them to familiarise themselves with the computer interface and the TDS methodology.
- Hutchings, S. C., Cha, W., Dunshea, F. R., Sharma, C., & Torrico, D. D. (2022). Understanding dominance: The effect of changing the definition of dominance when using TDS with consumers. Journal of Sensory Studies, e12750.
Q9. 2.7.2. Measurement of Emotions: why create a new scale and not use a validated scale with food-related emotions like the EsSense25 ?
Responses:
A new scale was not created. We have added following information in-text.
Please refer to page 6, line 252 – 263.
The CATA method involved the use of a check box for panellists to indicate their emotions. A study by Ng et al (2013) reported both check-all-that-apply (CATA) and EsSense yielded emotional data that clearly discriminated samples. Both EsSense and CATA data produced similar emotional spaces and product configurations. The consumer defined lexicon using CATA in fact provided a rich and more balanced list of positive and negative emotions specific to the product category. This is the reason why the CATA method was used to measure emotions.
- Ng, M., Chaya, C., & Hort, J. (2013). Beyond liking: Comparing the measurement of emotional response using EsSense Profile and consumer defined check-all-that-apply methodologies. Food Quality and Preference, 28(1), 193-205.
Results
Q10. Figure 2: Caption: “Differences in valence and arousal during..” should be “Mean valence and arousal ratings during” .
Responses:
The sentence has been corrected. We have changed it to “Mean valence and arousal ratings during”. Please refer to page 9, line 349 - 350.
Q11. What does the letters a,b, c refer to?
Responses:
We have clarified this. We have added “Different letter a, b, c indicates changes in mean affective ratings readings with different superscripts indicating significance (p < .05)." Please refer to page 9, line 350 - 352.
Q12. Figure 3: are these data for N=64? Do the curves represent the mean of all individual dominance curves of all participants?
Responses:
Yes, these data is for N=64. According to Pineau et al (2009), the curves represent the average dominance rates for all the panels. We have added a sentence that clarifies this in Figure 3.
Please refer to page 12, line 384 -385
- Pineau, N., Schlich, P., Cordelle, S., Mathonnière, C., Issanchou, S., Imbert, A., ... & Köster, E. (2009). Temporal Dominance of Sensations: Construction of the TDS curves and comparison with time–intensity. Food Quality and Preference, 20(6), 450-455.
Discussion
Q13. Section 4.3: is there a significant difference between LM and LMPS, or does LM simply dominate PS?
Responses:
We have actually carried the Cochran’s Q test to determine significance between sound conditions in terms of emotions. We have included the results in Appendix 4. In response to your questions (referring to Appendix 4):
- There was no significant differences between LM and LMPS in terms of positive emotions.
- LM evoked significantly higher positive emotions than PS.
We have explained this further. Please refer to page 16, line 507 - 509.
Using the Cochran's Q test (refer to Appendix 4.), there was no significant differences between LM and LMPS in terms of positive emotions. However, LM evoked significantly higher positive emotions than PS.
Q14. Appendix 3:
- Waterfall & jungle sounds is no longer available
- Burning fire does not play
Since sites like this one are so volatile it would be better if the authors could provide the set of stimuli for download as additional material to guarantee their availability.
Responses:
We have updated all the sound links on a google shared link.
Please refer to page 18 - 19, line 593, Appendix 2.
Reviewer 2 Report
The manuscript worked the influence of musical and non-musical sounds on the flavor perception of chocolate ice cream and emotion responses. The paper is interesting, and well written and presented. But there are still a problem need to explain.
As stated in introduction, a lot of literature has shown the significant influence of music on the flavor perception of foods and emotion responses, as well as the role of emotion on the perception of foods (such as ice cream, Xu et al., 2019). Why does the author still want to study the influence of music on the perception of chocolate ice cream and emotional responses? In addition, there are only two kinds of music in this study, liking (or pleasant) and disliking (or unpleasant), so could the positive or negative effect of music on the chocolate ice cream and emotional response be predicted? Therefore, the author should clearly state the importance or the novelty of this paper in the "introduction" and "objectives".
Some minor suggestions and questions:
Line 2 title, delete can, and replace "perception" by the "flavor perception"
Line 138, Line 143, Should it be "Appendix 1"?
Line 153, table 1 was not cited in the main text
Line 158, why the selected scale here is different with the that in 2.4.1(Line 147-148)
Line 317, add the α level.
Fig.2, add the y-axis in the figure 2
Table 2, and why the author selects the "0.707"? add the color explanation of "green and red" in the table.
Fig. 3, add the y-axis,
Line 347-348 why is the 14.29% and 21.48%? usually, 15% and 20%?
Line 418 why use the PLS-PM model in this study? The relationship of three dimentions shoud be further described.
Line 547 add the literature resource of term and description that in appendix 3, if any.
Author Response
Reviewer 2
The manuscript worked the influence of musical and non-musical sounds on the flavor perception of chocolate ice cream and emotion responses. The paper is interesting, and well written and presented. But there are still a problem need to explain.
Q1 As stated in introduction, a lot of literature has shown the significant influence of music on the flavor perception of foods and emotion responses, as well as the role of emotion on the perception of foods (such as ice cream, Xu et al., 2019). Why does the author still want to study the influence of music on the perception of chocolate ice cream and emotional responses? In addition, there are only two kinds of music in this study, liking (or pleasant) and disliking (or unpleasant), so could the positive or negative effect of music on the chocolate ice cream and emotional response be predicted? Therefore, the author should clearly state the importance or the novelty of this paper in the "introduction" and "objectives".
Responses:
Although music has been shown to influence flavour perception of foods, and emotional responses, no studies have been carried out to investigate how a mixture of musical and non-musical sounds varying in liking influenced flavour perception. In real life, both music and sound are experienced together.
We have reiterated the importance in the introduction and objectives.
Introduction: Please refer to page 2, line 58 - 60. However, the influence of a mixture of musical and non-musical sounds on temporal perception has not been investigated.
Objectives: Please also refer to page 2 - 3, line 93 - 95. The novelty of this study is to provide an understanding of how a mixture of musical and non-musical sounds varying in pleasantness can influence temporal changes in food perception.
Some minor suggestions and questions:
Q2. Line 2 title, delete can, and replace "perception" by the "flavor perception"
Responses:
The word “can” has been deleted. The word “perception” has been replaced to “flavour perception”. Please refer to page 1, line 2.
Q3. Line 138, Line 143, Should it be "Appendix 1"?
Responses:
Yes, we have made reference to “Appendix 1”. Please refer to page 3 - 4, line 133 - 144.
Panellists rated their liking of fourteen different music genres (Rentfrow & Gosling, 2003) as shown in Appendix 1. A 45-s segment of each genre was played. Studies by Kantono et al. [7-9] found that listening to music for 45 s was the optimum time for the rating of the pleasantness of ice cream. Music was selected and categorised for each genre ac-cording to the Apple iTunes music classification system. A preliminary trial was carried out with sixty people (21 to 40 years old) who regularly listened to music to identify the song that best represented each genre (see Appendix 1). The music samples were then modified using Adobe Audition CC version 11.1.1.3 (Adobe, California, USA) to scale the average sound pressure level (SPL) to 70 dB SPL. Music was played through a Sennheiser headset (Series HD 518: Sennheiser Electronics GmbH and Co. KG), and participants rated their liking of each genre (see Appendix 1) using a 100 mm unstructured line scale anchored with ‘extremely dislike’ and ‘extremely like’ at each end of the scale.
Q4. Line 153, table 1 was not cited in the main text
Responses:
“Table 1” has now been added in the main text in several places.
Please refer to section 2.5, page 5, line 194 - 195. The description and reference standards for these attributes are summarised in Table 1.
Please refer to section 2.5.1, page 5, line 214 - 215. The definition and references for each attribute is summarized in Table 1.
Q5. Line 158, why the selected scale here is different with the that in 2.4.1(Line 147-148)
Responses:
The scale used to measure sounds was according to Kantono et al (2016) who measured pleasantness of sounds using an unstructured 100 mm scale, anchored extremely unpleasant and extremely pleasant at each end.
- Kantono, K., Hamid, N., Shepherd, D., Lin, Y. H. T., Yakuncheva, S., Yoo, M. J., ... & Carr, B. T. (2016). The influence of auditory and visual stimuli on the pleasantness of chocolate gelati. Food quality and preference, 53, 9-18.
Q6. Line 317, add the α level. (p value)
Responses:
The α level (p value) has been added. Please refer to page 9, line 350 - 352.
Different letter a, b, c indicates changes in mean affective ratings readings with different superscripts indicating significance (p < 0.05).
Q7. Fig.2, add the y-axis in the figure 2
Responses:
The unit for y-axis has been added (cm) in Figure 2. Please refer to page 9, Figure 2.
Q8. Table 2, and why the author selects the "0.707"? add the color explanation of "green and red" in the table.
Responses:
We have added an explanation on why 0.707 was selected:
Please refer to page 9, line 354 - 361.
MDA was applied to determine the cosine values between −1 and +1 (Meyners et al, 2013) to show how sound conditions influenced emotion attributes when consuming chocolate ice cream. Table 2 highlights the emotions that are positively and negatively correlated with the combined auditory conditions using the first two dimensions of the CA bi-dimensional map. The relationship between emotional attributes and auditory conditions with cosine values > 0.707 are further explained [36]. This is because the au-thors reported that for interpretation, absolute cosines below 0.707 (=cos(45°) = −cos(135°)) indicate hardly any relationship at all.
The color explanations of “green and red” in the table has been added to the table. Please refer to page 10, Table 2.
Q9. Fig. 3, add the y-axis,
Responses:
The unit for y-axis has now been labelled with % as the dominance rate was converted into percentage. Please refer to page 11, Figure 3.
Q10. Line 347-348 why is the 14.29% and 21.48%? usually, 15% and 20%?
Responses:
An explanation has been given to address this comment.
Please refer to page 7, line 290 - 304.
The TDS time period is presented as standardized time (ST), and data are converted to percentages (0–100%; [35]). Time data was standardized to a score between 0 and 100 for each participant, where 0 represents the first click on the line scale, and 100 when the panellist either clicked stop or when the recording stopped automatically. The calculations of significant and chance levels were carried out according to Pineau et al. [24].
Chance level is the dominance rate that an attribute can obtain by chance. The value of chance level (P0) is equal to 1/p, p being the number of attributes, which in this case was seven. Therefore, the value of chance level is 14.29%. Significance level is the minimum value this proportion should be equal to be considered as significantly higher than P0. It is calculated using the confidence interval of a binomial proportion based on a normal approximation. The formula is shown below:
Where Ps is the lowest significant proportion value (α = 0.05) at any point in time for a TDS curve, n is the number of subject times replication (64 x1). Therefore, the value of significance level is 21.48%.
- Pineau, N., Schlich, P., Cordelle, S., Mathonnière, C., Issanchou, S., Imbert, A., ... & Köster, E. (2009). Temporal Dominance of Sensations: Construction of the TDS curves and comparison with time–intensity. Food Quality and Preference, 20(6), 450-455.
Q11. Line 418 why use the PLS-PM model in this study? The relationship of three dimentions shoud be further described.
Responses:
We have explained why PLS-PM was used in this study, and added in-text.
Please refer to page 14, line 456 - 466.
PLS-PM was used in this study to illustrate the emotional mechanism on how affective states and emotional responses can influence sensory perception as shown in Figure 5. PLS-PM is mainly used to construct cause-and-effect models. The use of PLS-PM emerged as an important tool to explain how the merging of food choice questionnaire and sensory perception influenced consumer behaviour of low sugar products (da Veiga et al., 2021), to demonstrate the relationships between physiological measurements, self-reported emotions, and sensory measurements when consuming chocolate gelato under different music conditions (Kantono et al., 2019), and for linking data about consumers, products and acceptance (Menichelli et al., 2014). One of the main advantages of PLS-PM is that it calculates reliability and validity at the same time (Ringle, Sarstedt, & Straub, 2012).
We have further improved the explanation on the relationship between the three dimensions of affective ratings, self-reported emotions and sensory perception.
Please refer to page 14, line 467 - 474
Valence was significantly positively correlated with positive emotions (pleasant, interested, joy, relaxed, active, at ease, calm, satisfied, excited, enthusiast, and energetic). Positive emotions were in turn positively correlated with sweetness, while showing negative correlations with bitterness and creaminess. Arousal on the other hand had significant positive correlations with negative emotions (annoyed, restless, tired, uneasy, un-happy, anxious, boredom, lonely, and unable to concentrate). Negative emotions were in turn positively correlated with bitterness and roasted, while being negatively correlated to sweetness, milky, creaminess, and cocoa.
- da Veiga, G. C., Johann, G., Lima, V. A., Kaushik, N., & Mitterer‐Daltoé, M. L. (2021). Food Choice Questionnaire and PLS‐Path modeling as tools to understand interest in low sugar products. Journal of Sensory Studies, 36(5), e12667.
- Kantono, K., Hamid, N., Shepherd, D., Lin, Y. H. T., Skiredj, S., & Carr, B. T. (2019). Emotional and electrophysiological measures correlate to flavour perception in the presence of music. Physiology & behavior, 199, 154-164.
- Menichelli, E., Hersleth, M., Almøy, T., & Næs, T. (2014). Alternative methods for combining information about products, consumers and consumers’ acceptance based on path modelling. Food quality and preference, 31, 142-155.
- Ringle, C. M., Sarstedt, M., & Straub, D. W. (2012). Editor's comments: a critical look at the use of PLS-SEM in" MIS Quarterly". MIS quarterly, iii-xiv.
Q12. Line 547 add the literature resource of term and description that in appendix 3, if any.
Responses:
No referenced literature source were used for this. We have used the following online websites to describe the emotional terms and provide examples:
https://www.collinsdictionary.com/dictionary/english
https://dictionary.cambridge.org/
Would you like us to include these websites in Appendix 3?
